# Dynamic Adsorption of As(V) onto the Porous α-Fe₂O₃/Fe₃O₄/C Composite Prepared with Bamboo Bio-Template

**Yuqing Peng** [1], **Yanhong Li** [2,*], **Shen Tang** [2], **Lihao Zhang** [2], **Jing Zhang** [1], **Yao Zhao** [1], **Xuehong Zhang** [3] **and Yinian Zhu** [3]

1   College of Environmental Science and Engineering, Guilin University of Technology, Guilin 541004, China; pengyuqing@glut.edu.cn (Y.P.); zjzj6662022@163.com (J.Z.); zhaoyao201509@163.com (Y.Z.)
2   Guangxi Key Laboratory of Environmental Pollution Control Theory and Technology, Guilin University of Technology, Guilin 541004, China; tangshen@glut.edu.cn (S.T.); lhzhang@glut.edu.cn (L.Z.)
3   Collaborative Innovation Center for Water Pollution Control and Water Safety in Karst Area, Guilin University of Technology, Guilin 541004, China; zhangxuehong@x263.net (X.Z.); zhuyinian@glut.edu.cn (Y.Z.)
*   Correspondence: lyh1685@163.com

**Abstract:** Arsenic (As(V)), a highly toxic metalloid, is known to contaminate wastewater and groundwater and is difficult to degrade in nature. However, the development of highly efficient adsorbents, at a low cost for use in practical applications, remains highly challenging. Thus, to investigate the As(V) adsorption mechanism, a novel porous α-Fe₂O₃/Fe₃O₄/C composite (PC-Fe/C-B) was prepared, using bamboo side shoots as a bio-template, and the breakthrough performance of the PC-Fe/C-B composite-packed fixed-bed column in As(V) removal was evaluated, using simulated wastewater. The PC-Fe/C-B material accurately retained the hierarchical porous microstructure of the bamboo bio-templates, and the results demonstrated the great potential of PC-Fe/C-B composite, as an effective adsorbent for removing As(V) from wastewater, under the optimal experimental conditions of: influent flow 5.136 mL/min, pH 3, As(V) concentration 20 mg/L, adsorbent particle size < 0.149 mm, adsorption temperature 35 °C, PC-Fe/C-B dose 0.5 g, and breakthrough time 50 min (184 BV), with $q_{e,exp}$ of 21.0 mg/g in the fixed-bed-column system. The CD-MUSIC model was effectively coupled with the transport model, using PHREEQC software, to simulate the reactive transportation of As(V) in the fixed-bed column and to predict the breakthrough curve for column adsorption.

**Keywords:** porous α-Fe₂O₃/Fe₃O₄/C composite; bamboo bio-template; arsenate; dynamic adsorption; CD-MUSIC model; PHREEQC

## 1. Introduction

The contamination of wastewater and groundwater with arsenate (As(V)) is one of the most concerning environmental issues globally [1,2]. Arsenic is a highly toxic metalloid that is released into the environment, mainly through human activities. It has heavy metal characteristics, in that, it is difficult to degrade in nature and undergoes bioaccumulation; it can enter the human body through the food chain, drinking water, respiration, and other ways, thereby posing serious dangers to human health. It is, also, listed as a group 1 carcinogen because of its high toxicity and carcinogenicity [3,4]. However, it, also, has certain non-metal characteristics. For example, As-containing compounds are unstable in water, readily decompose to amphoteric oxides, and easily migrate and transform in the environment [5,6].

In recent years, treatment of wastewater containing arsenic has received increasing attention. In this respect, Glass [7] modified polyacrylonitrile ultrafiltration membranes with amine functionalities, and the modified membrane showed that the arsenate adsorption capacity was five times higher than that of a pristine PAN membrane. Lee [8] developed a new beaded adsorbent, using coal-mine-drainage sludge, and this effectively removed arsenic

and other heavy metals from acid-mine drainage. Of all the As(V)-removal technologies, adsorption has been found to be the simplest and most effective method [9–11]. Adsorption has multiple advantages for use in water decontamination, including its low cost, high efficiency, and ease of operation as well as the absence of secondary-pollutant production. However, developing highly efficient adsorbents, at a low cost with good adaptability to packed-bed operation, for use in practical applications, remains highly challenging.

Previous studies have shown that effective As(V) adsorbents include iron oxide, aluminium oxide, clay, rare-earth oxides, and coal-fly ash. Among these, iron oxides (such as haematite, magnetite, goethite, akaganeite, feroxyhyte, siderite, and iron hydroxide) have been proven to be the best As(V) adsorbents [12–17]. Iron oxides can effectively immobilise As(V) through sorption processes because of their large surface areas and charged reactive surface sites, which provide extremely high sorption capacities [18,19]. By taking advantage of iron oxides and the hierarchical porous structure of Moso bamboo, a porous $\alpha$-$Fe_2O_3$/$Fe_3O_4$/C composite (PC-Fe/C-B) was prepared, using the side shoots of Moso bamboo as a microstructural template [20].

The surface-complexation model (SCM) has been employed to study the adsorption behaviour of As(V) on PC-Fe/C-B, and it is a vital tool used in studying the adsorption behaviour of ions. The charge-distribution-multi-site complexation (CD-MUSIC) model [21] is an advanced SCM for ion adsorption to metal oxides [22], and it is one of the most widely used models to describe As adsorption behaviour on metal-oxide surfaces. The CD-MUSIC model simultaneously considers the heterogeneity of the adsorption sites on the surface of the adsorbent, the spatial distribution of the charge of the adsorption centre atom in the electric double layer, and the contribution to the charge neutralisation of the oxygen atom in the adjacent electric layer. It, also, follows the three core principles of mass action, mass balance, and electrostatic interactions between reactants. A surface reaction equation system is constructed, and a chemical-equilibrium calculation is conducted under certain constraints. The simulation results can be used to predict the possible surface-complex forms after adsorption, such as inner or outer adsorption, monodentate or pluridentate surface complexes, protonation and protonation degree, and surface-charge change. The CD-MUSIC model is, mainly, based on actual experimental data, such as adsorption isotherms and pH, which are increasingly used in the study of adsorption processes at solid/liquid interfaces. In this respect, Salazar-Camacho and Villalobos (2010) [23] modelled arsenate adsorption on goethite, using experimental data from several previously published studies. Gustafsson (2001) [24] studied the adsorption of As(V) in soil, using the CD-MUSIC model, and found that oxide minerals (ferrihydrite and gibbsite) and clay minerals (proto-imogolite allophane) in the soil were the main adsorption interfaces. Furthermore, Liang et al. (2021) [25] revealed the complex species of Pb on the haematite surface, using CD-MUSIC modelling.

In this study, the PC-Fe/C-B adsorbent was prepared, using bamboo side shoots as a bio-template, which was systematically characterised using scanning-electron microscopy (SEM), energy-dispersive X-ray spectroscopy (EDS), Brunauer–Emmett–Teller (BET) surface-area analysis, X-ray diffraction (XRD), X-ray photoelectron spectroscopy (XPS), and Fourier-transform infrared (FT-IR) spectroscopy. The adsorption performance of PC-Fe/C-B was, also, evaluated, using a fixed-bed-column system with simulated As(V)-containing wastewater, and the adsorption kinetics, influencing factors, and adsorption mechanism of As(V) on the composites were systematically investigated. The adsorption data were, subsequently, analysed in detail, using several models, including the Thomas model, the Yoon–Nelson model, the Adams–Bohart model, the Clark model, and the Wolborska model. For further study of adsorption behaviour, CD-MUSIC modelling was conducted, using PHREEQC software (Version 3.0), which integrates CD-MUSIC with the transport block. PHREEQC [26] is a computer program used to simulate geochemical reactions and migration processes in natural or polluted water, and it performs complex speciation, intermittent reactions, one-dimensional (1D) transport, and geochemical-inversion simulations. As such, it has been widely applied in thermodynamic studies. Based on the

characterisation of the PC-Fe/C-B surface and the CD-MUSIC simulation, with the program PHREEQC, the speciation distribution and surface complexation on the PC-Fe/C-B adsorbents, during As(V) adsorption, were analysed to investigate the adsorption mechanism.

## 2. Materials and Methods

### 2.1. Preparation and Characterization

#### 2.1.1. Materials and Reagents

Moso bamboo was obtained from Guangxi, China. The reagents Fe $(NO_3)_3$, ethanol, $Na_3AsO_4 \cdot 12H_2O$, $HNO_3$, NaOH, and 5% ($v/v$) ammonia solutions were purchased from the Country Medicine Group, Shanghai, China. All reagent solutions were prepared from analytical-grade chemical reagents and ultrapure water.

#### 2.1.2. Sample Preparation

The side shoots of Moso bamboo were pre-treated, according to the method presented by Zhu et al. (2013) [20], and used as a template for preparing the composite adsorbent. In brief, the adsorbent PC-Fe/C-B was prepared by extraction pre-treatment, cleaning, drying, cyclic soaking in ferric nitrate solution, and roasting. Specifically, the following procedure was followed: (1) Chopped bamboo shoots were heated in 5% dilute ammonia, at 100 °C for 6 h, followed by washing with ultrapure water and drying at 80 °C for 24 h. (2) The pre-treated bamboo templates were immersed in a 1.2 mol/L precursor solution, at 60 °C, which was prepared by dissolving ferric nitrate in a mixture of ethanol and ultrapure water (1:1 volume ratio). The bamboo templates were soaked in the solution for 5 d, and the precursor solution was regularly checked to ensure even immersion. Subsequently, the samples were removed from the solution and oven-dried, at 85 °C for 1 d. The soak-dry process was repeated thrice. (3) The samples were carbonised in a muffle furnace, by slow heating to 600 °C and holding at this temperature for 3 h. After cooling to room temperature, samples with different particle sizes were obtained, by grinding and screening.

#### 2.1.3. Characterization of PC-Fe/C-B Sorbent

The specific surface area and porosity distribution of PC-Fe/C-B sorbent with different particle sizes ranging from 1.7–300.0 nm were measured by Quantachrome NOVAe1000, using a nitrogen adsorption–desorption method, with helium as the carrier gas and nitrogen as the adsorbate, at a liquid-nitrogen temperature of 77 K. All samples were degassed at 150 °C, in vacuum. The porosity of large pores (pore sizes of 0.0071–0.60 μm) was tested, using a mercury porosimeter (PoreMasterGT 60, Quantachrome, Boynton Beach, FL, USA). The test conditions were as follows: sample volume, 0.0977 g; mercury surface tension, 480 erg/cm$^2$; and mercury contact angle, 140°. The surface morphology of the PC-Fe/C-B sorbent was examined using SEM (JEOL JSM-6380LV, Japan Electron Optics Ltd., Mitaka, Tokyo), and its chemical composition was assessed using EDS. The as-prepared samples were characterised by XRD analysis (X'Pert PRO, PANalytical B.V., the Netherlands), using Cu K$\alpha$ radiation ($\lambda$ = 0.15406 nm). The samples were, also, measured in the form of KBr pellets, over the range 4000–400 cm$^{-1}$, using FT-IR spectroscopy (Nicolet Nexus 470, Thermo Fisher Scientific Inc.). XPS measurements, using an X-ray photoelectron spectrometer (VG ESCALab Mark II spectrometer; VG Scientific Ltd., East Grinstead, UK), were conducted on samples before and after adsorption.

### 2.2. Arsenate Adsorption to PC-Fe/C-B

Simulated wastewater containing As(V) was prepared, using $Na_3AsO_4 \cdot 12H_2O$ and $HNO_3$. Dynamic-adsorption experiments were performed, using a small fixed-bed-column adsorption device. The fixed-bed column had a length of 200 mm, with outer and inner diameters of 35 mm and 12 mm, respectively. The concentration of As(V) in the aqueous solution was analysed, by an AAS flame and graphite furnace atomic absorption spectrophotometer (AA700, Perkin Elmer, Boston, MA, USA).

The factors influencing the dynamic adsorption of As(V) were investigated, by changing the initial As(V) concentration of the influent (10 mg/L, 20 mg/L, 30 mg/L, and 50 mg/L), influent flow rate (4.28 mL/min, 5.136 mL/min, 6.849 mL/min, and 8.561 mL/min), influent pH (1.0, 2.0, 3.0, 4.0, 5.0, 6.0, and 7.0), adsorbent dose (0.3 g, 0.4 g, 0.5 g, and 0.6 g), particle size of the adsorbent (0.841–0.4 mm, 0.4–0.25 mm, 0.25–0.177 mm, 0.177–0.149 mm, and <0.149 mm), and adsorption temperature (25 °C, 35 °C, and 45 °C). To obtain the main dynamic adsorption-process parameters and to determine the best model to describe the dynamic adsorption behaviour, the adsorption behaviour was analysed, using several models, including the Thomas model [27], the Nelson model [28,29], the Bohart model [30], the Clark model [31], and the Wolborska model [32].

*2.3. Modeling*

The CD-MUSIC model is based on the SCM approach, developed by Vaughan Jr. et al. [33,34]. It can be used to describe the adsorbent surface-acid–base reactions and adsorbate–surface reactions that constitute chemical equilibrium, and it was, thus, selected to model the surface adsorption behaviour in this study. Simulations and parameter optimisation were performed, using PHREEQC (Version 3.0). The complete CD-MUSIC model incorporates a basic Stern-layer approach, iron-oxide-surface characteristics, As(V)-aqueous-phase speciation, surface-acid–base reactions, and adsorbate–surface reactions.

Two data blocks of PHREEQC were used to achieve the coupled simulations for As(V) migration–adsorption [26], which were advection and transport. In the transport data block of this paper, an adsorption column composed of 10 cells was simulated with a total amount of adsorbent weighing 0.5 g = (10 × 0.05 g), total adsorption column height of 1.23 cm = (10 × 0.123 cm), and inflow rate of Q = 5.136 mL/min. The water-chemistry parameters, PC-Fe/C-B-surface properties, aqueous-phase As speciation, transport-parameter modelling, and surface complexation are described in detail in the PHREEQC input file and the explanation notes in the electronic supplementary material (Tables S6–S9).

## 3. Results and Discussion

### 3.1. Characterization of PC-Fe/C-B

The BET method was used to determine the specific surface area of PC-Fe/C-B, with different particle sizes at 77 K. The BET-specific surface areas were determined as ranging from 138.17 m$^2$/g to 186.66 m$^2$/g, with corresponding pore volumes ranging from 0.07487 cm$^3$/g to 0.11938 cm$^3$/g, which suggests that PC-Fe/C-B exhibited good adsorption potential. Under the test range of pore sizes ranging from 1–300 nm, most of the PC-Fe/C-B pores ranged from 2.0–19.0 nm (74–81.3%)—within the mesopore range, although some were within the micropore range of <2 nm (18.3–26%). The results indicated that PC-Fe/C-B was a hierarchically structured porous material with a medium pore size. The large pore size and porosity of blocky PC-Fe/C-B were determined, using the mercury injection method. PC-Fe/C-B was found to contain abundant pores and to have a porosity of 69.51%.

The SEM results, also, showed that PC-Fe/C-B preserved the hierarchical porous structure of the bamboo wood (Figure 1). Three different types of pores (macropores, mesopores, and micropores) were retained from the bio-templates, and these originated from vessels (widths 50–70 μm), parenchyma cells (10–30 μm), fibre pores (widths 3–10 μm) and pits on the walls of the vessels and parenchyma cells (widths 0.1–1.3 μm), respectively. It is known that interpore connectivity can be significantly increased by extraction, pre-treatment, and high-temperature calcination. Quantitative SEM–EDS peak-area analysis showed that up to 4.19% (*w/w*) As(V) was adsorbed on the PC-Fe/C-B material (Figure 1). EDS spectra were collected, from samples treated with an As(V) concentration of 100 mg/L.

The XRD results from samples, before and after As(V) adsorption, are shown in Figure 2, indicating that magnetite, hematite, and carbon were present on the PC-Fe/C-B surface. In the preparation process, the cellulose in the bamboo template went through the processes of dehydrogenation and deoxidation by high-temperature calcination, and

the adsorbed-ferric-nitrate precursor chemically decomposed into iron oxide ($\alpha$-Fe$_2$O$_3$). In a high-temperature oxygen-containing environment, part of the iron oxide obtained from decomposition was further transformed into magnetic iron oxide (Fe$_3$O$_4$). Therefore, the crystal forms of iron oxide in PC-Fe/C-B were Fe$_3$O$_4$ and $\alpha$-Fe$_2$O$_3$. It can be seen from Figure 2 that the characteristic peaks of $\alpha$-Fe$_2$O$_3$, at $2\theta$ = 24.50°, 40.85°, 49.53°, and 63.98°, disappeared after As(V) adsorption; the main characteristic peaks for $\alpha$-Fe$_2$O$_3$ at $2\theta$ of 33.21° and 54.12° were significantly weakened; and the characteristic peaks for Fe$_3$O$_4$ at $2\theta$ of 29.98°, 43.34°, 57.50°, and 62.42° were strengthened, which indicated that the crystal form of iron oxide changed after adsorption. It was speculated that this may be due to the hydration reaction of some $\alpha$-Fe$_2$O$_3$ dissolved in water, to generate FeOOH or a surface-complexation reaction.

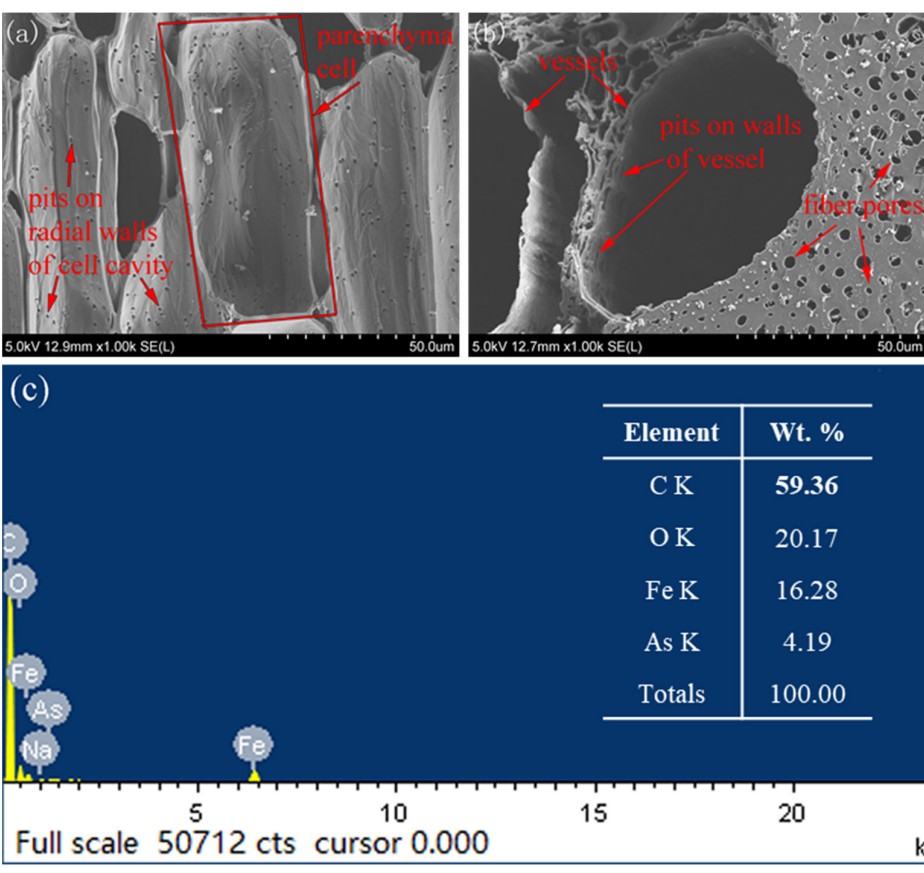

**Figure 1.** SEM images of PC-Fe/C-B, after As(V) adsorption: (**a**) transverse section. (**b**) longitudinal section and (**c**) EDS spectrum of PC-Fe/C-B, after As(V) adsorption.

In the FT-IR spectra (Figure 3) of PC-Fe/C-B after As(V) adsorption, the presence of a wide absorption peak at 3415.26 cm$^{-1}$ was attributed to the contracting vibration of $-$OH in the material, rather than to the dissociation of physically adsorbed water (due to the absence of a vibration peak near 1630 cm$^{-1}$) [35]. This, also, showed that the number of $-$OH groups on the surface of iron oxide increased after As(V) adsorption, which, also, confirmed the acid–base dissociation behaviour of PC-Fe/C-B, during the adsorption of As. After As adsorption, the intensity of the stretching vibration peak of CO$_2$ at 2361 cm$^{-1}$ decreased, and the C-O and C=O vibration peaks [35], at 1541 cm$^{-1}$ and 1516 cm$^{-1}$, shifted to higher wave numbers, of 1551 cm$^{-1}$ and 1532 cm$^{-1}$, respectively. The Fe-O vibration peaks, at 564 cm$^{-1}$ and 443 cm$^{-1}$, also, shifted to lower wavenumbers, of 556 cm$^{-1}$ and 440 cm$^{-1}$, indicating that the groups corresponding to these peaks were the active adsorption sites on the surface of PC-Fe/C-B. In the FT-IR spectra of the PC-Fe/C-B sorbent after sorption of As(V), peaks of AsO$_4$$^{3-}$ for As–O were observed to be bending around 873.60 cm$^{-1}$.

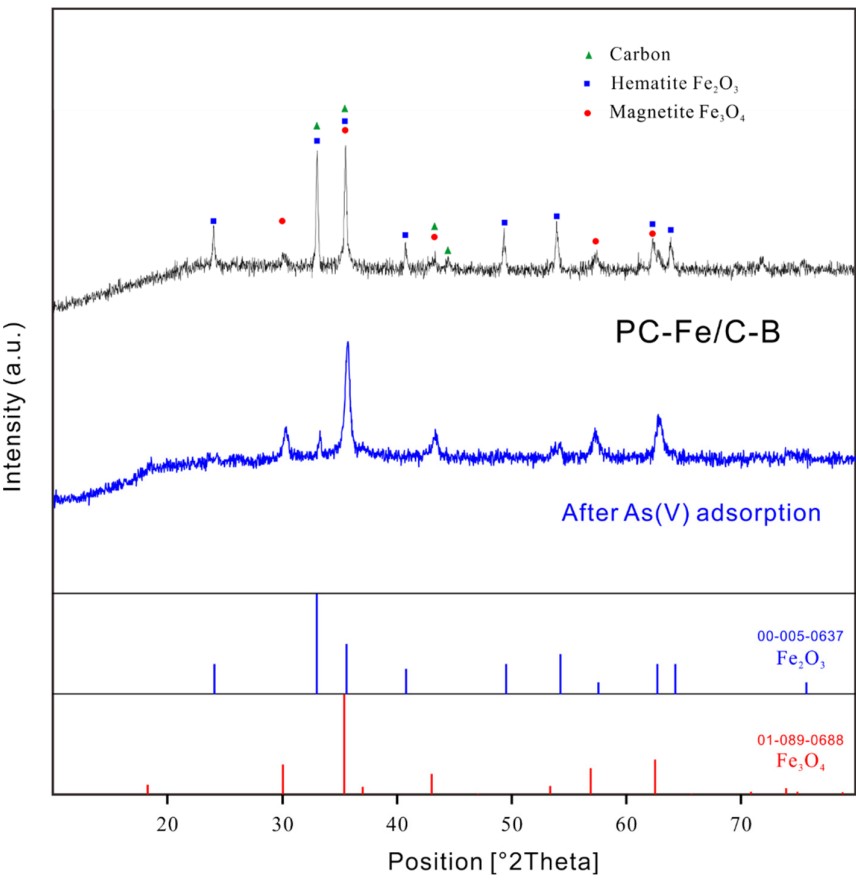

**Figure 2.** XRD patterns of PC-Fe/C-B, before and after adsorption.

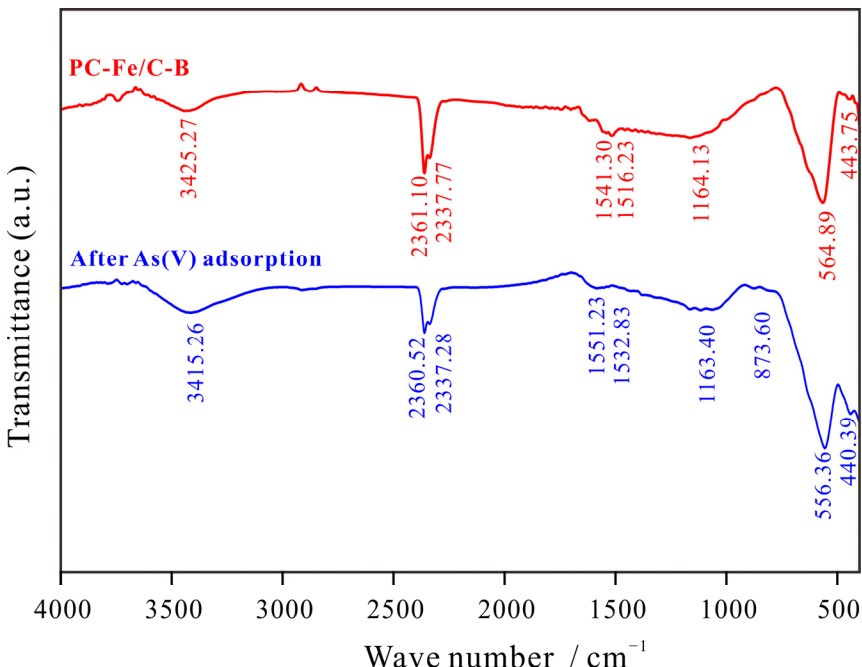

**Figure 3.** FT-IR patterns of PC-Fe/C-B, before and after As(V) adsorption.

The XPS survey spectra and high-resolution Fe 2p, C 1s, and O 1s spectra of PC-Fe/C-B, after As(V) adsorption, are shown in Figure 4. After As(V) adsorption, the As 3d peak, with a binding energy of 42.78 eV, was detected, which indicated that As existed as As(V) without a redox reaction [36]. The O 1s and Fe 2p peaks, after As(V) adsorption by

PC-Fe/C-B, showed almost no shift, indicating that the chemical states of O and Fe did not change after As(V) adsorption. The atomic percentages of PC-Fe/C-B, after As adsorption, were 58.96% C, 28.72% O, 11.07% Fe, and 1.25% As. Moreover, since the charge amount of the above electron transfer was far less than a unit of electricity, it was speculated that the transfer of the outer shared electron pair occurred during the bonding process.

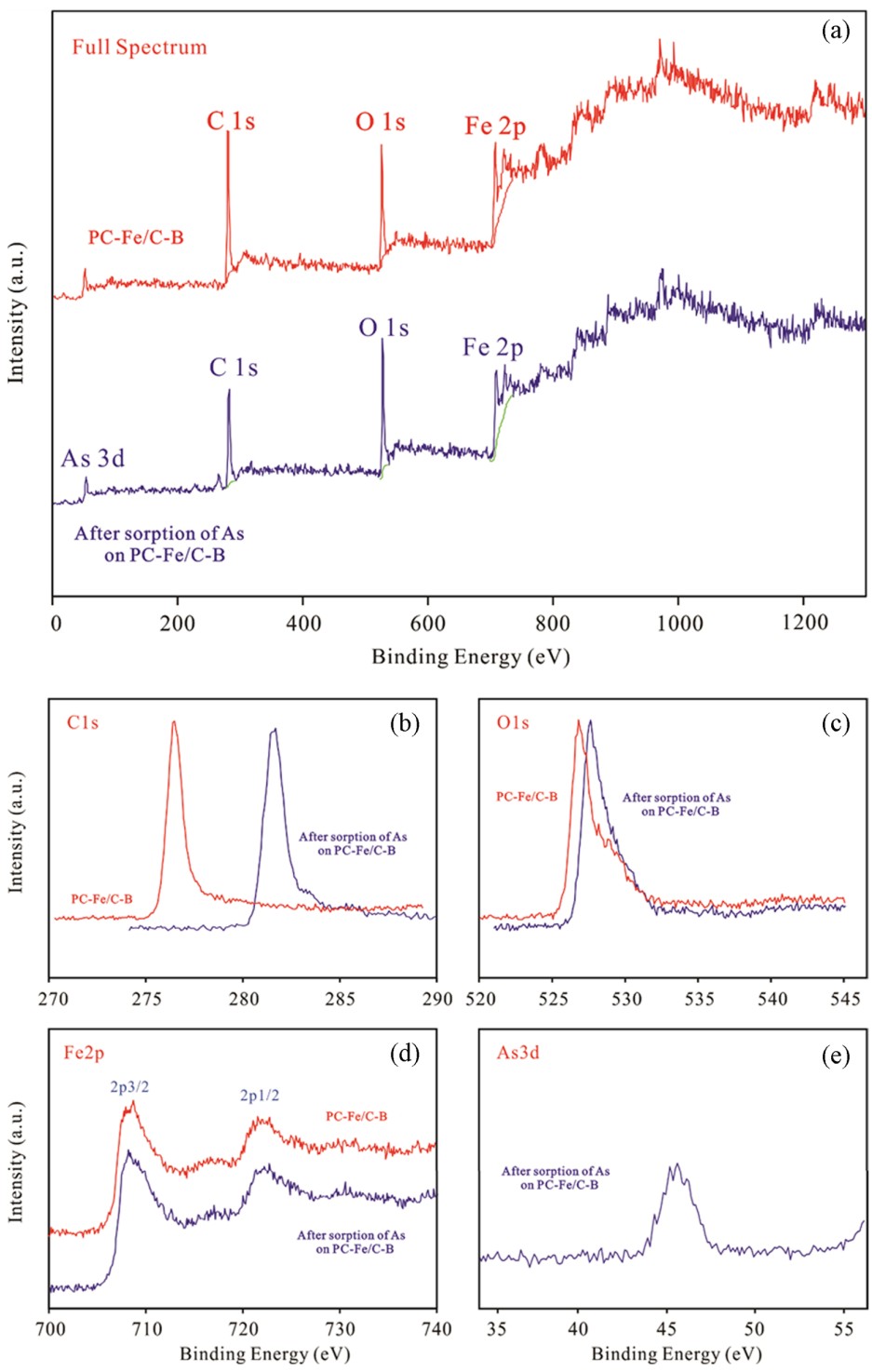

**Figure 4.** (**a**) XPS survey spectra and high-resolution. (**b**) C 1s. (**c**) O 1s. (**d**) Fe 2p and (**e**) As 3d spectra of PC-Fe/C-B, before and after As(V) adsorption.

### 3.2. As(V) Dynamic Adsorption

The results showed that the PC-Fe/C-B adsorbent could be used to remove As(V) effectively from wastewater, under the following optimal experimental conditions: influent flow 5.136 mL/min, pH 3, As(V) concentration 20 mg/L, adsorbent particle size < 0.149 mm, adsorption temperature 35 °C, PC-Fe/C-B dose 0.5 g, and breakthrough time 50 min (184 BV), with $q_{e,exp}$ of 21.0 mg/g. Increasing the PC-Fe/C-B adsorbent amount (adsorption-column height) and reducing the particle size, effectively prolonged both the breakthrough time and failure time as well as improved the quality of the column, thus improving the performance of the adsorption column. In addition, the total As(V) removal rate increased with increasing PC-Fe/C-B dose. However, increasing the influent concentration and flow rate accelerated column failure. PH had a significant influence on As(V) adsorption, and acidic conditions were conducive to adsorption (pH 2–4). Temperature had little effect on the adsorption of As(V) by PC-Fe/C-B. The effects of each factor on As(V) adsorption are shown in Figures 5–10.

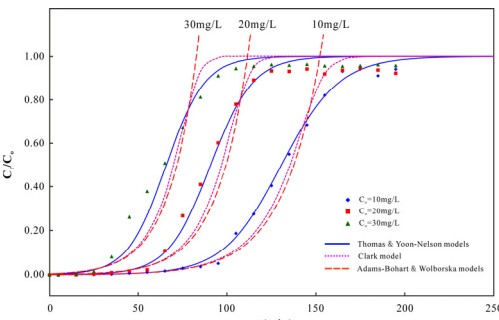

**Figure 5.** Effect of influent concentration on breakthrough curve of PC-Fe/C-B adsorption As(V).

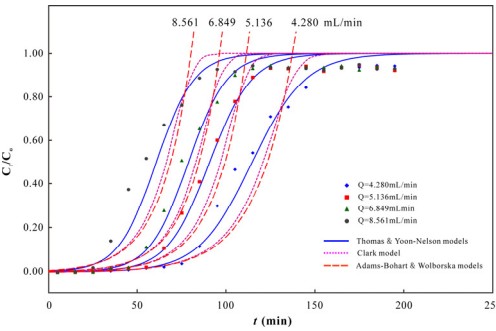

**Figure 6.** Effect of influent flow rate on breakthrough curve of PC-Fe/C-B adsorption As(V).

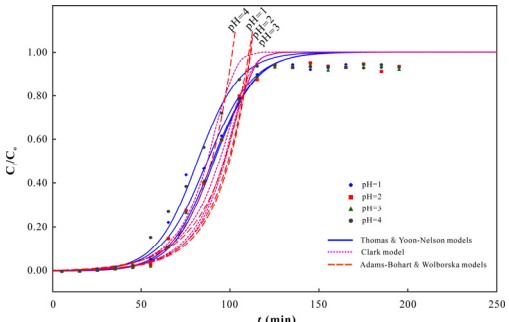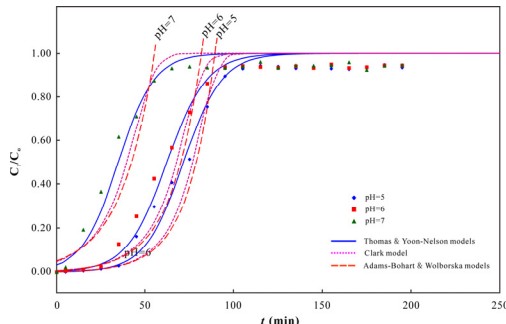

**Figure 7.** Effect of influent pH on breakthrough curve of PC-Fe/C-B adsorption As(V).

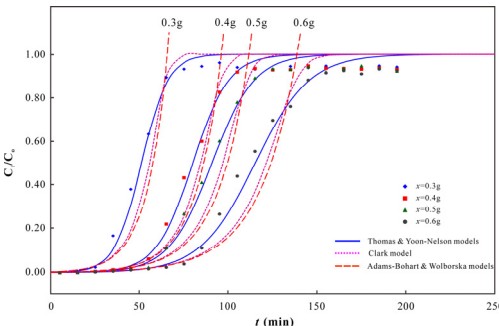

**Figure 8.** Effect of adsorbent dose on breakthrough curve of PC-Fe/C-B adsorption As(V).

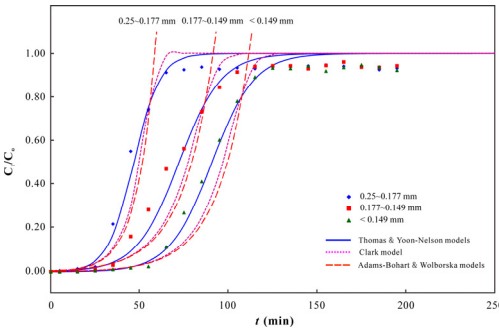

**Figure 9.** Effect of adsorbent grain size on breakthrough curve of PC-Fe/C-B adsorption As(V).

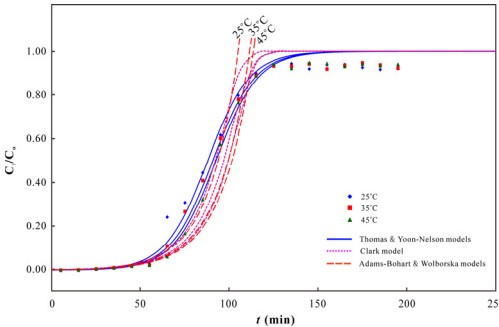

**Figure 10.** Effect of temperature on breakthrough curve of PC-Fe/C-B adsorption As(V).

*3.3. As(V) Adsorption Capacity of PC-Fe/C-B*

Table S5 in the electronic Supplementary Material compares the As(V) adsorption capacities of PC-Fe/C-B with various adsorbents reported in the literature. As shown in Table S5, PC-Fe/C-B has a great potential for the adsorption and removal of As(V) from solution. PC-Fe/C-B has a larger adsorption capacity than several adsorbents tested under similar conditions that investigated particle size and initial concentration. Some of these adsorbents are as follows: an aluminium mining by-product (4.51 mg/g) [37], PBGC-Fe/C (4.83 mg/g) [20], organo-modified natural zeolite materials (6.7 mg/g) [38], biochar-magnetite composite (5.49 mg/g) [14], Chinese red soil (0.936 mg/g) [39], iron-coated seaweeds (7.3 mg/g) [40], goethite-P (AAm) composite (1.22 mg/g) [13], iron-oxide-based adsorbents μGFH (22.4 mg/g) and μTMF (15.4 mg/g) [41], Fe-sericite composite beads (5.78 mg/g) [42], and red mud-modified biochar (5.923 mg/g) [43].

The adsorption capacity of PC-Fe/C-B was, thus, found to be 21.0 mg/g comparable to, but lower than, the following adsorbents: aluminium-modified crop-straw-derived biochars (34.71–52.02 mg/g) [44], zeo-NaY-S (37.59 mg/g) [45], magnetic nanocomposite (250 mg/g) [46], mesoporous-sulphated zirconia (99.23 mg/g) [47], polystyrene-$Fe_3O_4$ (139.3 mg/g) [48], and potassium ferrate (162.02 mg/g) [12]. In general, the smaller the particle size is, the larger the specific surface area and the higher the adsorption capacity.

However, most adsorbents with large adsorption capacities are nanoparticles, and these are both unsuitable for fixed-bed-column adsorption and difficult to recycle. Moreover, nanoparticles easily agglomerate. PC-Fe/C-B can be easily ground into particles of a demanding size that are more suitable than nanoparticles for use in fixed-bed columns, for adsorptive As(V) removal [49]. Furthermore, most of the reported adsorption capacities were based on the results of static adsorption experiments. Compared with static adsorption [20], the diffusion condition in the flow state is superior in dynamic adsorption, and this enables full contact between the arsenate and adsorbent in the solution. Therefore, a dynamic experiment was more useful than a batch experiment, for evaluating the adsorption performance of the adsorbent. Similar to a previous study [50], the higher adsorption capacity of PC-Fe/C-B was attributed to the longer contact time between the adsorption material and the solution as well as better water-conservancy-diffusion conditions.

### 3.4. CD-MUSIC Modeling

3.4.1. Simulation Parameters of CD-MUSIC Model with PHREEQC

The CD-MUSIC model is an SCM developed from the classical TLM model, for the complexation of heavy-metal ions in hydrated oxides. Iron oxide combines metals and protons at both strong and weak sites, and the charges, potentials, and types of adsorbates are distributed in a double-layer plane, of more than two layers; therefore, its mass transfer region extends from the surface of the adsorbent to the free solution (Figure 11). There are three major differences between the CD-MUSIC and TLM models: the representation of surface acidity, the placement of ions and charges in the electrostatic plane, and the representation of reactive surface adsorption sites [51]. In the CD-MUSIC model, protonated surface groups are located in the 0-plane, charges of specific adsorbed ions (such as arsenate) are distributed between the 0-plane and 1-plane, and supporting electrolyte ions are assumed to be single-point charges located in the 1-plane. The diffuse layer starts from the 2-plane and extends to the bulk solution. $C_1$ and $C_2$ are the key parameters in the model, and these are 0.85 F/m$^2$ and 0.75 F/m$^2$, respectively [18].

CD-MUSIC is by far the most flexible model. However, it has one limitation, which is that many parameters need to be determined, and they are not equally constrained [52]. However, the CD-MUSIC model accurately characterised the surface-charge characteristics of PC-Fe/C-B, and the surface-charge characteristic parameters obtained by fitting are listed in Tables 1 and 2. These are both the PC-Fe/C-B properties and the basic parameters for the adsorption-complex-reaction simulation.

**Table 1.** Speciation of As(V) in aqueous phase.

| As(V) Aqueous-Phase-Dissociation Reaction | logK |
|:---:|:---:|
| $H_3AsO_4 = AsO_4^{3-} + 3H^+$ | −20.7 |
| $H^+ + AsO_4^{3-} = HAsO_4^{2-}$ | 11.5 |
| $2\,H^+ + AsO_4^{3-} = H_2AsO_4^-$ | 18.46 |

According to a study by Hiemstra et al. (1996) [21], octahedrally coordinated iron-oxide surfaces mainly contain three types of surface sites: singly ($\equiv$FeO), doubly ($\equiv$Fe$_2$O), and triply ($\equiv$Fe$_3$O) coordinated sites, as shown in Figure 12. There are three types of Fe-O surface hydroxyl sites for iron oxides, namely $\equiv$FeOH$_2^+$, $\equiv$FeOH, and $\equiv$FeO$^-$, but the unit-bound oxygen $\equiv$FeOH$_2^+$ in the outer layer is, mainly, complexed with adsorbate-functional groups, such as As-O [21,53,54]. Combination forms include monodentate, bidentate, and tridentate combinations. The surface protonation reaction is defined by Equations (1) and (2), and the model parameters and ion-pair formation reactions for the singly coordinated groups are listed in Table 2.

**Table 2.** Surface-complexation-equilibrium reactions of As(V) adsorption on PC-Fe/C-B, and the parameters used in CD-MUSIC model simulation.

| Surface Reaction of Adsorption of $AsO_4^{3-}$ by PC-Fe/C-B | logK |
|---|---|
| $\equiv FeOH = \equiv FeO^- + H^+$ | $-8.93$ |
| $\equiv FeOH + H^+ = \equiv FeOH_2^+$ | 7.29 |
| $2 \equiv FeOH + 2 H^+ + AsO_4^{3-} = \equiv Fe_2O_2AsO_2^- + 2 H_2O$ | 29.29 |
| $2 \equiv FeOH + 3 H^+ + AsO_4^{3-} = \equiv Fe_2O_2AsOOH + 2 H_2O$ | 32.69 |
| $\equiv FeOH + 2 H^+ + AsO_4^{3-} = \equiv FeOAsO_2OH^- + H_2O$ | 26.62 |
| The surface site density of PC-Fe/C-B/(sites/nm$^2$): 2.5 | |

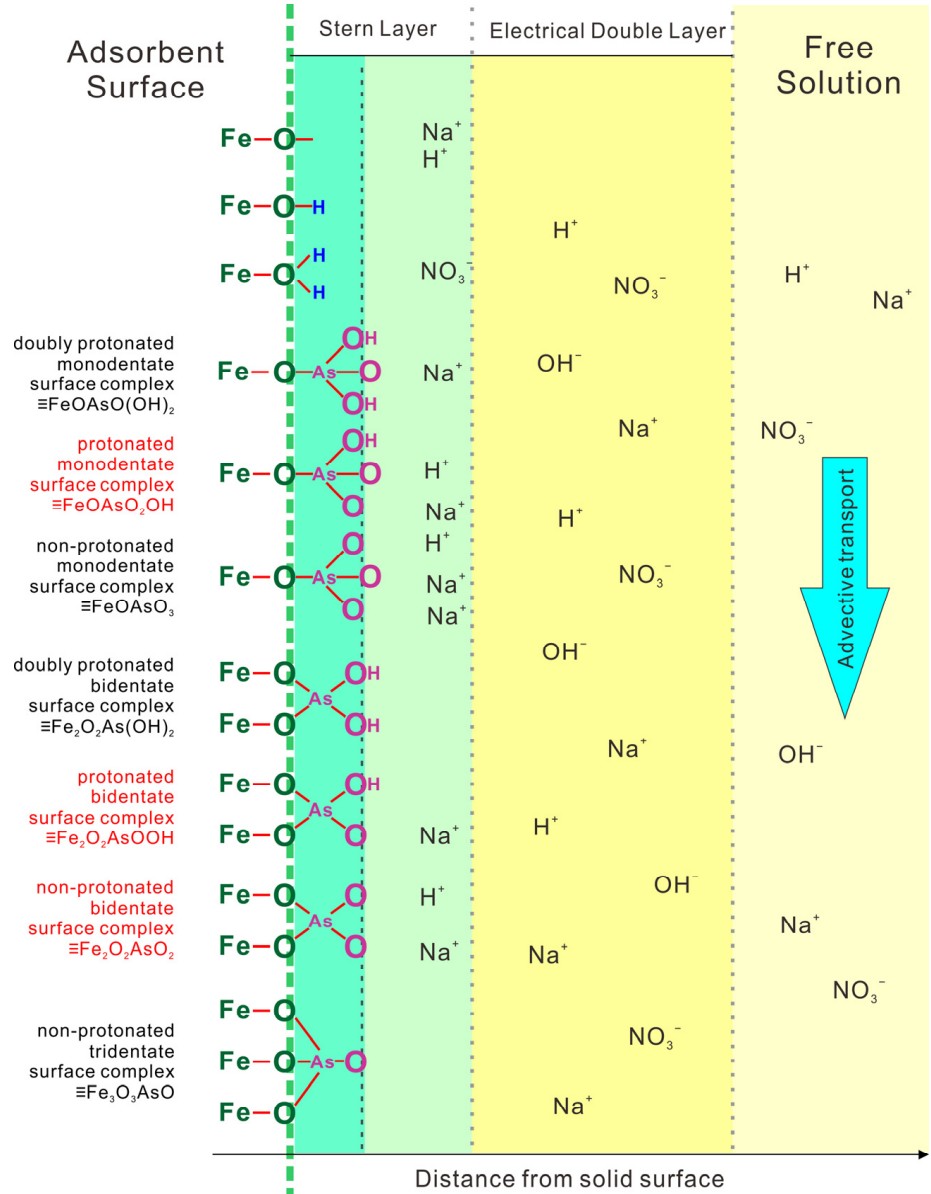

**Figure 11.** Schematic representation of the parameters of the CD-MUSIC adsorption model.

$$\equiv FeOH_2^+ == \equiv FeOH + H^+ \qquad (1)$$

$$K_1 = \left[ \equiv FeOH \right]\left[ H^+ \right] / \left[ \equiv FeOH_2^+ \right] \qquad (2)$$

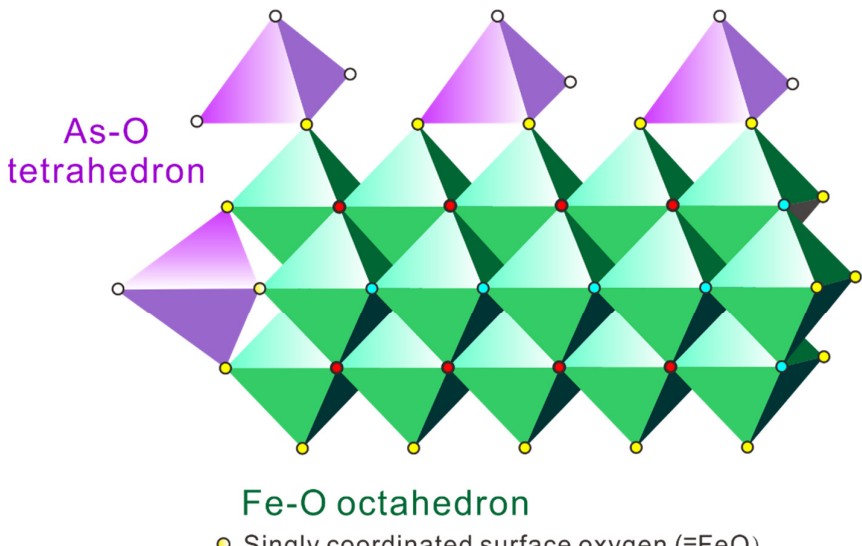

**Figure 12.** Charge on singly, doubly, and triply coordinated-surface hydroxyl -functional groups, in octahedrally coordinated iron oxide.

It is usually considered that bidentate-surface complexes are inferior to monodentate complexes, for systems with a low As(V) coating coverage. However, Hayes et al. (1988) [55] reported simulation results for monodentate and bidentate surface species that were almost identical, and, thus, simpler surface species (monodentate) were used in this study to model As(V) adsorption [33,34].

The existing aqueous phase As(V) forms can be obtained from the following dissociation equations, and there are four possible As(V) surface reactions, in the formation of the above four aqueous As(V) species.

$$H^+ + AsO_4^{3-} \Leftrightarrow HAsO_4^{2-} \tag{3}$$

$$2H^+ + AsO_4^{3-} \Leftrightarrow H_2AsO_4^- \tag{4}$$

$$3H^+ + AsO_4^{3-} \Leftrightarrow H_3AsO_4^0 \tag{5}$$

Of the four possible aqueous As(V) species, only two ($H_2AsO_4^-$ and $HAsO_4^{2-}$) exist at typical pH values for drinking-water treatment (pH = 4–10). Therefore, in the SCF model simulation, it is necessary to consider the surface reactions given in Equations (9) and (11) (please see the equations provided below). When modelling the As(V)-FeAC system with the SCF model [33,34], the adsorption balance described using Equations (9) and (11) is referred to as the 2-RXN model. The 3-RXN model considers the effects of three surface reactions, when describing the adsorption balance, and Equation (7) (rather than Equation (13)) was selected as the third reaction (again, please see equations listed below). The removal rate of As(V) in a low-pH environment is generally the highest [56]. Although as As(V) is exclusively adsorbed on the oxide surface, by a ligand-exchange mechanism, and exists as a complex on the inner-sphere surface [33,34], four As(V) surface reactions are possible in the aqueous phase.

The column adsorption results were simulated by a transport–adsorptive reaction, using the PHREEQC program, and the surface-site density of PC-Fe/C-B was 2.50 sites/nm$^2$. Xie et al. [57] studied the adsorption of Cr(VI) on goethite, using the CD-MUSIC model, and estimated the site density to be 2.7 sites/nm$^2$, which is close to our estimation.

$$XOH^0 + H_3AsO_4 \Leftrightarrow XH_2AsO_4^0 + H_2O \tag{6}$$

$$K_{As1} = \frac{\left[XH_2AsO_4^0\right]}{\left[XOH^0\right]\{H_b^+\}\left[AsO_4^{3-}\right]} \exp\left(\frac{3F\Psi}{RT}\right) \tag{7}$$

$$K_{As2} = \frac{\left[XHAsO_4^-\right]}{\left[XOH^0\right]\{H_b^+\}\left[AsO_4^{3-}\right]} \exp\left(\frac{2F\Psi}{RT}\right) \tag{8}$$

$$XOH^0 + HAsO_4^{2-} \Leftrightarrow XAsO_4^- + H_2O \tag{9}$$

$$K_{As3} = \frac{\left[XAsO_4^{2-}\right]}{\left[XOH^0\right]\{H_b^+\}\left[AsO_4^{3-}\right]} \exp\left(\frac{F\Psi}{RT}\right) \tag{10}$$

$$XOH^0 + AsO_4^{3-} \Leftrightarrow XOHAsO_4^{3-} \tag{11}$$

$$K_{As4} = \frac{\left[XOHAsO_4^{3-}\right]}{\left[XOH^0\right]\left[AsO_4^{3-}\right]} \tag{12}$$

As mentioned above, the possible surface-complexation forms of PC-Fe/C-B-adsorbed As(V) included monodentate mononuclear, bidentate binuclear, and tridentate trinuclear. The adsorption reactions for As(V) in the CD-MUSIC-model simulation involved the following reactions: a surface-protonation reaction of the adsorption material $\equiv FeOH_2^+$, a combination reaction for two electrolyte-ion pairs on the site of $\equiv FeOH_2^+$, an arsenic acid three-stage dissociation reaction, and an adsorption reaction of $AsO_4^{3-}$ on the surface of PC-Fe/C-B. The basic parameters in Tables 1 and 2 as well as the adsorption data under different conditions were entered into the software, to simulate the CD-MUSIC model. The f value (CD factor) of each surface morphology was manually adjusted, and the macro adsorption data were used as the constraint conditions of the model, by trial calculation, until appropriate results were obtained, as shown in Figure 13. The possible complex forms of $AsO_4^{3-}$ adsorbed by PC-Fe/C-B include the doubly protonated monodentate-surface complex $\equiv FeOAsO(OH)_2$, protonated monodentate-surface complex $\equiv FeOAsO_2OH$, non-protonated monodentate-surface complex $\equiv FeOAsO_3$, doubly protonated bidentate-surface complex $\equiv Fe_2O_2As(OH)_2$, protonated bidentate-surface complex $\equiv Fe_2O_2AsOOH$, non-protonated monodentate-surface complex $\equiv Fe_2O_2AsO_2$, and non-protonated tridentate-surface complex $\equiv Fe_3O_3AsO$.

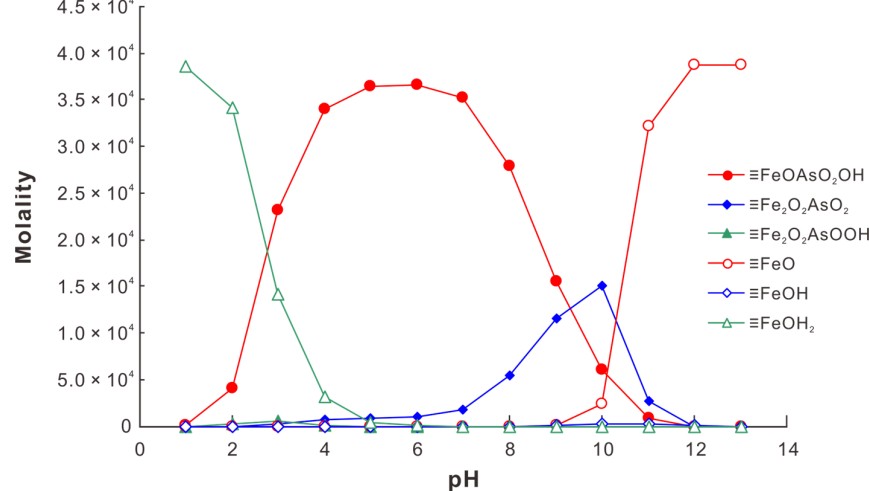

**Figure 13.** Speciation distribution of surface complexation on PC-Fe/C-B, after As(V) adsorption (initial As = 20 mg/L).

### 3.4.2. Simulation Calculation of Solid-Surface-Complexing State

The adsorption behaviour of As(V) on PC-Fe/C-B at pH values of 1–13 was simulated using the CD-MUSIC model, and the changing trends of various forms under different pH conditions were predicted, as shown in Figure 13. The PHREEQC input files are detailed in the electronic Supplementary Material.

The simulation results in Figure 13 show that the possible complex forms of $AsO_4^{3-}$ adsorbed by PC-Fe/C-B were protonated monodentate-surface complex $\equiv FeOAsO_2OH$ (complex constant of 26.62), protonated bidentate-surface complex $\equiv Fe_2O_2AsOOH$ (complex constant of 32.69) and non-protonated monodentate-surface complex $\equiv Fe_2O_2AsO_2$ (complex constant of 29.29). In a wide range of pH, the dominant form was protonated monodentate-surface complex $\equiv FeOAsO_2OH$, with the highest concentration at pH 4–7. At pH 9–12, the non-protonated monodentate-surface complex $\equiv Fe_2O_2AsO_2$ began to predominate. At pH > 10, the main form of the surface oxide was $\equiv FeO$, which explained the low adsorption capacity of the adsorbent, under high-pH conditions.

### 3.4.3. Simulation of Reactive Transportation in Dynamic-Adsorption Process Using PHREEQC

As previously described, As adsorption has been extensively studied, using the CD-MUSIC model. Recently, Cui (2015) [58] used granular $TiO_2$ to remove arsenic from groundwater in geogenic areas, and the As breakthrough curve was successfully calculated, using PHREEQC, incorporating the CD-MUSIC model and the 1D reactive transport block. In this study, the CD-MUSIC model was used to fit the experimental data of fixed-bed-column adsorption, and the fitting data were compared with those of the aforementioned classical dynamic model (see Figure 14). As shown in Figure 14, reactive transportation was simulated and calculated using the PHREEQC-coupled CD-MUSIC-surface-complexation model, of As(V) adsorbed by PC-Fe/C-B with the transport model of As(V) in aqueous solution. The breakthrough curve in the column-dynamic-adsorption process fitted well with the predicted data. In addition, the correlation coefficient ($R^2$) was higher than that of the previous Thomas, Yoon–Nelson, Adams–Bohart, Clark, and Wolborska models. This satisfactory modelling result proves the effective application of CD-MUSIC, for predicting the column-breakthrough curves of As(V) adsorption in arsenic wastewater.

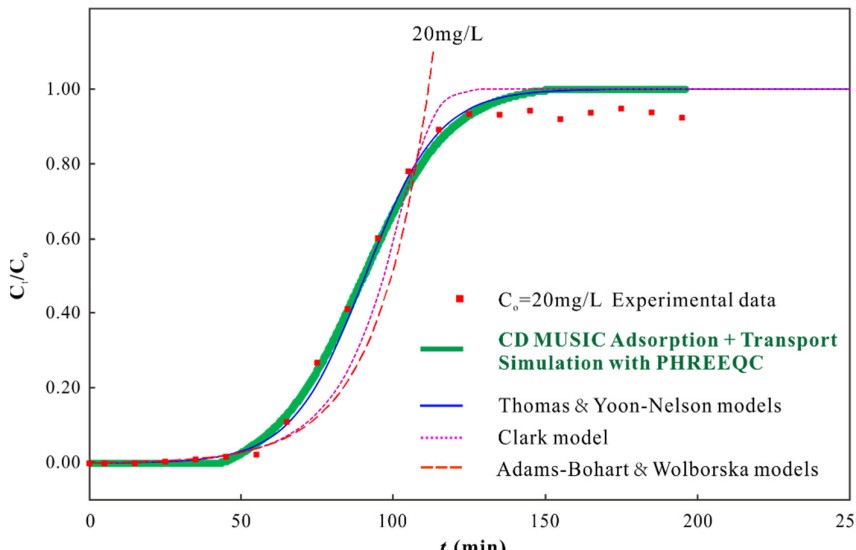

**Figure 14.** Experimental data and coupled CD-MUSIC–transport simulation, using PHREEQC, for PC-Fe/C-B adsorption As(V) (initial concentration of As(V) = 20 mg/L, pH = 3, Q = 5.136 mL/min, dose m = 0.5 g, particle size < 0.149 mm, T = 35 °C).

## 4. Conclusions

In this study, PC-Fe/C-B material was prepared, and the hierarchical porous microstructure of bamboo bio-templates was found to be accurately retained. The PC-Fe/C-B adsorbent was found to be effective for the removal of As(V) from wastewater, under the following optimal experimental conditions: influent flow 5.136 mL/min, pH 3, As(V) concentration 20 mg/L, adsorbent particle size 0.149 mm, adsorption temperature 35 °C, PC-Fe/C-B dose 0.5 g, and breakthrough time 50 min (184 BV), with $q_{e,exp}$ of 21.0 mg/g. The adsorption capacity of PC-Fe/C-B was higher than that of similar adsorbents reported in the literature.

The surface characteristics of PC-Fe/C-B were simulated and analysed, using the CD-MUSIC model with the PHREEQC program. The surface-site density of PC-Fe/C-B was estimated as 2.5 sites/nm$^2$, and the surface complexes ≡FeOAsO$_2$OH, ≡Fe$_2$O$_2$AsOOH, ≡Fe$_2$O$_2$AsO$_2$, and ≡FeOAsO$_2$OH were dominant over a wide pH range.

These results showed that the CD-MUSIC model can be effectively coupled with the transport model, using the PHREEQC program, to simulate the reactive transportation of As(V) in the fixed-bed column and to predict the breakthrough curve for column adsorption. The fitting result obtained was superior to that of classical models, such as the Thomas model and the Yoon–Nelson model.

**Supplementary Materials:** The following supporting information can be downloaded at: https://www.mdpi.com/article/10.3390/w14121848/s1. Table S1: Parameters of the Thomas model and the equilibrium As(V) uptake ($q_e$) and the total removal percentage (Y) for As(V) adsorption onto PC-Fe/C-B under different conditions; Table S2: Parameters of the Yoon—Nelson model, using linear-regression analysis for As(V) adsorption onto PC-Fe/C-B, under different conditions; Table S3: Parameters of the Clark model, using linear-regression analysis for As(V) adsorption onto PC-Fe/C-B, under different conditions; Table S4: Parameters of the Adams–Bohart and Wolborska models, using linear regression analysis for As(V) adsorption onto PC-Fe/C-B, under different conditions; Table S5: Comparison of the adsorption capacity of the PC-Fe/C-B and various adsorbents for As(V) removal from water; Table S6: Input files for the As(V) aqueous-phase speciation in the simulation, using PHREEQC; Table S7: PHREEQC input files for the surface acid–base behaviour of PC-Fe/C-B, in aqueous As(V) solution; Table S8: An input file example of the CD-MUSIC model for surface complexation in the simulation, using PHREEQC; Table S9: An input file example for the influent TRANSPORT in the simulation, using PHREEQC.

**Author Contributions:** Conceptualization, Y.Z. (Yinian Zhu) and Y.L.; methodology, Y.P. and L.Z.; software, Y.Z. (Yinian Zhu) and Y.L.; validation, Y.P.; formal analysis, J.Z.; investigation, Y.Z. (Yao Zhao); resources, X.Z. and Y.L.; data curation, S.T.; writing–original draft preparation, Y.P.; writ-ing–review and editing, Y.L.; visualization, L.Z.; supervision, S.T.; project administration, X.Z.; funding acquisition, Y.Z. (Yinian Zhu) and X.Z. All authors have read and agreed to the published version of the manuscript.

**Funding:** This work was financially supported by the National Science Foundation of China (No. 52070050) and the Natural Science Foundation of Guangxi (No. 2020GXNSFAA159017).

**Institutional Review Board Statement:** Not applicable.

**Informed Consent Statement:** Not applicable.

**Data Availability Statement:** The data used in this study are available on request from the corresponding author.

**Acknowledgments:** The authors thank Guilin University of Technology for providing the materials, equipment, and laboratory facilities required to successfully conclude this research. All authors would like to sincerely thank the anonymous reviewers for the improvement of this paper, through their constructive and insightful comments.

**Conflicts of Interest:** The authors declare no conflict of interest.

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
