# Peer review of "Dynamic Adsorption of As(V) onto the Porous α-Fe2O3/Fe3O4/C Composite Prepared with Bamboo Bio-Template"

_water, doi:10.3390/w14121848_

Round 1

Reviewer 1 Report

In this study, the authors investigated the adsorption behaviour of porous α-Fe2O3/Fe3O4/C Composite which was made using bamboo to remove As(V) from the waster water in the bed column.

The manuscript was written very well and well-composed,  and I can see well designed experimental system and produced results. However, the manuscript needs to be checked before being accepted.

Please see the following to improve the manuscript:

  • I think it will be better to use mm or micron sizes instead of mesh.
  • Please don't use personal “we” or “our” in the manuscript.
  • Some figures are coming before the corresponding text such as Figures 1, 2, and 3.
  • Please provide higher quality for Figure 3.
  • Also, there needs to be done the same for Figures 5 to 10. Hard to see the details in the figures.
  • How did you calculate the adsorption capacity? What formula was used?
  • You say that “the adsorption capacity is negatively correlated with the particle size”. Do you mean the surface area? Additional explanation welcome!!!
  • Also, you say that “moreover, nanoparticles are easy to agglomerate”. Do you mean that Nanoparticles have high adsorption capacity due to their high surface areas but they are more compacted in the column because of their sizes, hence low the adsorption capacity or longer time? What do you think?
  • Dynamic adsorption? Can it be used “continuous”?
  • Finally, there are some mistakes in the writing of the paper.
  • Please check the attached file for my corrections/comments.

Besides this, the manuscript can be accepted after the minor correction/revision to be published in the journal of Water.

Author Response

Response to reviewer 1 comments (water-1722410)

First of all, we greatly appreciate the comments and insightful suggestions from anonymous reviewers, and their time spent on our manuscript. With these suggestions, our manuscript was improved substantially. The following are our response to comments, point by point. We have made the modification in the revised manuscript (use the “Track Changes” function of MS Word, otherwise the line number will be wrong).

Comment: 1. I think it will be better to use mm or micron sizes instead of mesh.

Response: The relevant words have been already changed in mm in the revised manuscript with the revision mode.

Comment: 2. Please don't use personal “we” or “our” in the manuscript.

Response: The relevant words have been already changed in the revised manuscript with the revision mode. (Lines 63-64 and line 173)

Comment: 3. Some figures are coming before the corresponding text such as Figures 1, 2, and 3.

Response: The figures have been corrected in the revision. (Pages 6, 8, and 9)

Comment: 4. Please provide higher quality for Figure 3.

Response: The graph has been modified. (Page 9)

Comment: 5. Also, there needs to be done the same for Figures 5 to 10. Hard to see the details in the figures.

Response: The graphs have been redrawn in the revised manuscript. (Pages 11 and 12)

Comment: 6. How did you calculate the adsorption capacity? What formula was used?

Response: The Adsorption capacity was calculated by the following formula:

qe = (Q/1000∫(T0→T2) Cad dt)/x

where:

qe = equilibrium adsorption capacity of adsorbent for As(V) (mg/g).

Q = flow rate (mL/min).

T2 = adsorption saturation time (min), T0 =initial time (min).

Cad = concentration of adsorbate on adsorbent (mg/L).

x = adsorbent dosage (g).

Initial concentration of As(V) was 20 mg/L. Regular sampling and analysis of As(V) was taken with the effluent. Then Cad can be calculated with the concentration of As(V) in the influent and effluent solution.

Comment: 7. You say that “the adsorption capacity is negatively correlated with the particle size”. Do you mean the surface area? Additional explanation welcome!!!

Response: We mean the specific surface area. Generally speaking, the smaller the particle size, the larger the specific surface area, and the higher adsorption capacity.

The adsorption capacity is described versus particle sizes.

Comment: 8. Also, you say that “moreover, nanoparticles are easy to agglomerate”. Do you mean that Nanoparticles have high adsorption capacity due to their high surface areas but they are more compacted in the column because of their sizes, hence low the adsorption capacity or longer time? What do you think?

Response: Due to the small particle size, large proportion of surface atoms, specific surface area, and surface energy, nanoparticles are in an unstable state of energy. Therefore, nanoparticles are easy to agglomerate. Then formed secondary particles to increase the granular particle size.

Comment: 9. Dynamic adsorption? Can it be used “continuous”?

Response: Yes, it can be used “continuous”. The adsorption experiments were performed under dynamic conditions (continuous flow) in a small fixed-bed column adsorption device.

Comment: 10. Finally, there are some mistakes in the writing of the paper.

Response: We have carefully revised these mistakes.

Comment: 11. Please check the attached file for my corrections/comments.

Response: We deeply appreciate the reviewer’s suggestion and the relevant words have been already changed in the revised manuscript with the revision mode.

Reviewer 2 Report

The present article focuses on the adsorption of As(V) by a porous α-Fe2O3/Fe3O4/C composite elaborated with bamboo bio-template. The material was characterized by BET, SEM, FTIR, XRD, and XPS. The theme is current and can give visibility to the paper attracting the journal readership. However, the paper still needs some precision. I made the comments below as suggestions to improve the quality of the manuscript.

  1. The authors should go deeper in the discussion about the proposed composition of their materials. Why have they chosen hematite? Why not maghemite (it presents much more important magnetic properties)? Why not only magnetite?

  1. The authors state that hematite provides the main active sites for the As(V) since it participated in the chemical reactions during the adsorption process. What kind of reactions? For me, it could be considered a drawback if the reactions endow the surface sites unavailable for further adsorption.

  1. According to Table 2, the values of equilibrium constants for As(V) adsorption are too high (log K is at least 26). Would it not be unfavorable to reuse the proposed nanomaterials? Would it not be difficult for As(V) desorption? Authors should comment on this.

  1. In the introduction, the authors should emphasize that the ability of iron oxides to adsorb arsenic is related to As(V) (arsenate) rather than As(III) (arsenide). Even though the authors correctly specified that in the title, they should make it clear throughout the manuscript.

  1. This survey is only a laboratory study, with very simplified systems. Thus, I suggest to the authors a study with some real (or synthetic) wastewater and/or with some representative interferents to improve the visibility of the paper. For example, sulfate and phosphate are important interferents for As(V) adsorption.

  1. Concerning the FTIR spectra, the authors include in their discussion the band at 873.60 cm-1, which can be attributed to As-O stretching vibrations.

Author Response

Response to reviewer 2 comments (water-1722410)

First of all, we greatly appreciate the comments and insightful suggestions from anonymous reviewers, and their time spent on our manuscript. With these suggestions, our manuscript was improved substantially. The following are our response to comments, point by point. We have made the modification in the revised manuscript (use the “Track Changes” function of MS Word, otherwise the line number will be wrong).

Comment: 1. The authors should go deeper in the discussion about the proposed composition of their materials. Why have they chosen hematite? Why not maghemite (it presents much more important magnetic properties)? Why not only magnetite?

Response: See subsection 2.1.2 for the preparation method. After high temperature calcination, the adsorbed precursor solution of ferric nitrate was chemically decomposed into iron oxide (α-Fe2O3). In the high temperature oxygen-containing environment, part of the iron oxide obtained from decomposition was further transformed into magnetic iron oxide (Fe3O4). By extraction pretreatment, cleaning, drying, cyclic soaking in ferric nitrate solution, and roasting, the iron oxides (α-Fe2O3, Fe3O4) were loaded on the PC-Fe/C-B. Moreover, the XRD analysis results indicated that α-Fe2O3 was the main active adsorption phase.

Accept the suggestions of the reviewer and more discussion about the proposed composition of their materials has been added in the revision manuscript:

“Therefore, the crystal forms of iron oxide on PC-Fe/C-B were Fe3O4 and α-Fe2O3.The analysis showed that the cellulose in the bamboo template went through the process of dehydrogenation and deoxidation by high temperature calcination. The adsorbed precursor solution of ferric nitrate was chemically decomposed into iron oxide (α-Fe2O3). In the high temperature oxygen-containing environment, part of the iron oxide obtained from decomposition was further transformed into magnetic iron oxide (Fe3O4).” (Page 7, Lines 225-231)

Comment: 2. The authors state that hematite provides the main active sites for the As(V) since it participated in the chemical reactions during the adsorption process. What kind of reactions? For me, it could be considered a drawback if the reactions endow the surface sites unavailable for further adsorption.

Response: The reactions can be seen in the Table 2.

2 ≡FeOH + 2 H+ + AsO43- = ≡Fe2O2AsO2- + 2 H2O; logK = 29.29

2 ≡FeOH + 3 H+ + AsO43- = ≡Fe2O2AsOOH + 2 H2O; logK = 32.69

≡FeOH + 2 H+ + AsO43- = ≡FeOAsO2OH- + H2O; logK = 26.62

The surface site density of PC-Fe/C-B simulated by Transport-Adsorptive reaction with the PHREEQC program was 2.50 sites/nm2. High site density means high proton active surface sites and high adsorption capacity for As(V). And the surface site density met the requirements of adsorbing a certain amount of As(V).

Comment: 3. According to Table 2, the values of equilibrium constants for As(V) adsorption are too high (log K is at least 26). Would it not be unfavorable to reuse the proposed nanomaterials? Would it not be difficult for As(V) desorption? Authors should comment on this.

Response: In fact, we have done desorption experiments with NaOH, NaHCO3, NaCl, and water as desorption solutions as shown in the Figures 1 and 2 below. Under the conditions of 0.1 mol/L NaOH solution, flow rate of 5.14 mL/min, and desorption temperature of 35°C, the desorption rate of adsorbent was 66.3 %. Use of NaOH solution had good desorption effect. The relevant research has not been officially published.

Figure 1. Effect of desorption solvent types on the As(V) desorption efficiency

Figure 2. Effect of NaOH eluent concentration on the As(V) desorption efficiency

Comment: 4. In the introduction, the authors should emphasize that the ability of iron oxides to adsorb arsenic is related to As(V) (arsenate) rather than As(III) (arsenide). Even though the authors correctly specified that in the title, they should make it clear throughout the manuscript.

Response: The relevant words have been already changed in the revised manuscript with the revision mode.

Comment: 5. This survey is only a laboratory study, with very simplified systems. Thus, I suggest to the authors a study with some real (or synthetic) wastewater and/or with some representative interferents to improve the visibility of the paper. For example, sulfate and phosphate are important interferents for As(V) adsorption.

Response: Thank you for your precious comment and advice. In the next study, we will study the competitive adsorption of sulfate and phosphate with arsenate on the composition.

Comment: 6. Concerning the FTIR spectra, the authors include in their discussion the band at 873.60 cm-1, which can be attributed to As-O stretching vibrations.

Response: Accept and we have added the following to the discussion:

“In the FT-IR spectra of the PC-Fe/C-B sorbent after sorption of As(V), the peaks of AsO43− for the As–O bending around 873.60 cm−1 were observed.” (Page 9, Lines 255-256)

Reviewer 3 Report

Dear Authors,

The article is fatally worded and should never be published.

With kind regards,

Author Response

Many thanks for your help in dealing with our manuscript. The language presentation was improved with assistance from a native English speaker with appropriate research background. We sincerely hope the reviewers can give more details in order to improve our research. 

Reviewer 4 Report

The presented manuscript includes the study of the Dynamic Adsorption of As(V) onto the Porous α-Fe2O3/Fe3O4/C Composite Prepared with Bamboo Bio-template.

The results of the work are presented on a good level and well written, but, some general corrections should be mentioned.

  1. For the dynamic experiment it would be better to show the velocity of influent flow in mm/min in column through adsorbent, but not in ml/min. Unfortunately in MDPI, I can’t suggest exact nice papers as an example. Try to find something about modified activated carbon for deironing of underground water
  2. Please split subsection 2.1 into two: 2.1 Materials and reagents, and 2.2 Samples preparation
  3. Please mention all used reagents in subsection 2.1 Materials and reagents
  4. Instead of spectra fig.1c would be more useful to show EDS results in wt.%
  5. Please show the As phases in XRD for spent adsorbent
  6. Also would be great to present more information on BET analysis.

The manuscript needs minor revisions, but some new analysis is strongly recommended.

Author Response

Response to reviewer 4 comments (water-1722410)

First of all, we greatly appreciate the comments and insightful suggestions from anonymous reviewers, and their time spent on our manuscript. With these suggestions, our manuscript was improved substantially. The following are our response to comments, point by point. We have made the modification in the revised manuscript (use the “Track Changes” function of MS Word, otherwise the line number will be wrong).

Comment: 1. For the dynamic experiment it would be better to show the velocity of influent flow in mm/min in column through adsorbent, but not in ml/min. Unfortunately in MDPI, I can’t suggest exact nice papers as an example. Try to find something about modified activated carbon for deironing of underground water.

Response: This is mainly because the velocity of influent flow in the simulation using PHREEQC was mL/min. And the unit of mL/min is also convenient for us to compare our research with others. For instance, Lee Yonghyeon, et al. "Arsenic adsorption study in acid mine drainage using fixed bed column by novel beaded adsorbent." Chemosphere 291 (2022): 132894. (https://doi.org/10.1016/j.chemosphere.2021.132894)

Comment: 2. Please split subsection 2.1 into two: 2.1 Materials and reagents, and 2.2 Samples preparation.

Response: Subsection 2.1 has been split into three: 2.1.1. Materials and reagents, 2.1.2. Samples preparation, and 2.1.3. Characterization. (Page 3)

Comment: 3. Please mention all used reagents in subsection 2.1 Materials and reagents.

Response: All used reagents have been added. (Page 3)

Comment: 4. Instead of spectra fig.1c would be more useful to show EDS results in wt.%.

Response: We have added the results in wt.%. (Page 6)

Comment: 5. Please show the As phases in XRD for spent adsorbent.

Response: No obvious diffraction peaks corresponding to As were observed in the XRD pattern. Therefore, we did not show As phases in XRD for spent adsorbent.

Comment: 6. Also would be great to present more information on BET analysis.

Response: More information on BET analysis has been added as follow:

“The specific surface area and porosity distribution of PC-Fe/C-B sorbent with different particle sizes ranging from 1.7 nm to 300.0 nm were measured with a Quantachrome NOVAe1000, which was determined by nitrogen adsorption-desorption method with helium as carrier gas, nitrogen as adsorbate, and liquid nitrogen temperature of 77 K. All samples were degassed at 150°C in vacuum. For large pores with pore size of 0.0071 ~ 0.60 μm, the porosity was tested by a mercury porosimeter (PoreMasterGT 60, Quantachrome, USA). The test conditions were as follows: sample volume 0.0977 g, mercury surface tension 480 erg/cm2, mercury contact angle 140°.” (Page 3, Lines 129-136)

“BET method was used to determine the specific surface area of PC-Fe/C-B with different particle sizes at 77 K. The BET specific surface areas were determined to be 138.17~186.66 m2/g with the corresponding pore volume being 0.07487~0.11938 cm3/g, which suggested that PC-Fe/C-B had good adsorption potential. Under the test range of 1~300 nm pore size, most of the pores of PC-Fe/C-B were in the range of 2.0–19.0 nm (74–81.3%), belonging to the mesopore range. There was also a small part in the micropore range of < 2nm (18.3~26%). The result indicated that PC-Fe/C-B was a hierarchical-structured porous material with medium pore size. The large pore size and porosity of the blocky PC-Fe/C-B were measured by mercury injection method. The PC-Fe/C-B contained abundant pores with a porosity of 69.51%.” (Page 4, Lines 184-193)

Reviewer 5 Report

The authors in their manuscript described the performance iron oxide-based adsorbents for the removal of As(V) in water through small column adsorption tests and modelling of breakthrough curves using surface complexation model. They concluded that CD-MUSIC model is an was effectively coupled with the transport model to simulate the reactive transportation of As(V) in the fixed-bed column and to predict the breakthrough curve for the column adsorption. The narrative is well written although in places the language could be tightened up somewhat. This work is suitable for publication in Water. However, revisions are some suggestions for this study.

  1. Page 2, paragraph 1: Not only column adsorption but adsorptive membranes with positive polymers have been gained significant attention in the recent past for adsorption of As(V). See instance, Chemie Ingenieur Technik, 93(9), 1396-1400.
  2. Page 2, paragraph 1: Different minerals of iron oxides such as feroxyhyte and akaganeite are also well-known for arsenic removal. See, for instance, Journal of hazardous materials 400(2020), 123221.
  3. It is suggested to compare the adsorption capacity of the synthesized adsorbed (q= 21 mg/g) determined through dynamic adsorption in small-scale adsorption columns. See, for instance, Journal of Chemical Technology & Biotechnology, 96(6) (2022) for the adsorption capacities of some of the highly effective adsorbents. Comparison will be beneficial to highlight the adsorption performance of synthesized adsorbents over conventional adsorbents.

Author Response

Response to reviewer 5 comments (water-1722410)

First of all, we greatly appreciate the comments and insightful suggestions from anonymous reviewers, and their time spent on our manuscript. With these suggestions, our manuscript was improved substantially. The following are our response to comments, point by point. We have made the modification in the revised manuscript (use the “Track Changes” function of MS Word, otherwise the line number will be wrong).

Comment: 1. Page 2, paragraph 1: Not only column adsorption but adsorptive membranes with positive polymers have been gained significant attention in the recent past for adsorption of As(V). See instance, Chemie Ingenieur Technik, 93(9), 1396-1400.

Response: Accept the suggestions and we cite the paper to illustrate recent concerns. (Page 2, Lines 46-50)

Comment: 2. Page 2, paragraph 1: Different minerals of iron oxides such as feroxyhyte and akaganeite are also well-known for arsenic removal. See, for instance, Journal of hazardous materials 400(2020), 123221.

Response: We agree with the comment and we have added some other iron oxides such as akaganeite, feroxyhyte, and siderite. Also, the corresponding literatures have been cited. (Page 2, Line 58)

Comment: 3. It is suggested to compare the adsorption capacity of the synthesized adsorbed (q= 21 mg/g) determined through dynamic adsorption in small-scale adsorption columns. See, for instance, Journal of Chemical Technology & Biotechnology, 96(6) (2022) for the adsorption capacities of some of the highly effective adsorbents. Comparison will be beneficial to highlight the adsorption performance of synthesized adsorbents over conventional adsorbents.

Response: We are not sure whether you referred to this article. See Usman, M.; Belkasmi, A.I.; Kastoyiannis, I.A.; Ernst, M. Pre-deposited dynamic membrane adsorber formed of microscale conventional iron oxide-based adsorbents to remove arsenic from water: application study and mathematical modeling. J. Chem. Technol. Biotechnol. 2021, 96(6), 1504-1514. (https://doi.org/10.1002/jctb.6728)

In the study, the As(V) adsorption capacities of powdered-sized μGFH and μTMF were estimated to be 22.4 μg As(V) mg−1 and 15.4 μg As(V) mg−1, respectively, which are comparable or lower than that of the PC-Fe/C-B (q = 21 mg/g). These iron oxide-based adsorbents including PC-Fe/C-B can effectively remove As(V) from As(V)-polluted water. And the μGFH and μTMF have been cited in the comparison in subsection 3.3. (Page 12, Lines 304-305)

Round 2

Reviewer 1 Report

I see the authors have responded to the comments successfully and revised the manuscript very well.

Now, it can be accepted for publication in the journal of Water.

Author Response

We sincerely thank the anonymous reviewers for the improvement of this paper through their constructive and insightful comments.

Reviewer 2 Report

In this revised version, the authors improved the quality of the manuscript according to the reviewers' comments. Now, I recommend its publication.

Author Response

(The authors gave the same response as above.)

Reviewer 3 Report

Dear Authors,

The interpretation of the XRD results is incorrect and requires clarification and independent analysis (do not follow the interpretations in other articles too much). the conclusions are too general and should be refined.

Author Response

Response to reviewer 3 comments (water-1722410)

First of all, we greatly appreciate the comments and insightful suggestions from anonymous reviewers, and their time spent on our manuscript. Those comments are valuable and very helpful. The language presentation has been improved again by professional editors at Editage, a division of Cactus Communications (Please see the attachment for the editing certificate). The following are our response to comments, point by point. We have made the modification in the revised manuscript (use the “Track Changes” function of MS Word, otherwise the line number will be wrong).

Comment: 1. The interpretation of the XRD results is incorrect and requires clarification and independent analysis (do not follow the interpretations in other articles too much).

Response: Agreed with the reviewer's opinion and the sentence is rephrased, it now reads:

“The XRD results from samples before and after As(V) adsorption are shown in Figure 2, indicating that magnetite, hematite, and carbon were present on the PC-Fe/C-B surface. In the preparation process, the cellulose in the bamboo template went through the processes of dehydrogenation and deoxidation by high temperature calcination, and the adsorbed ferric nitrate precursor chemically decomposed into iron oxide (α-Fe2O3). In a high-temperature oxygen-containing environment, part of the iron oxide obtained from decomposition was further transformed into magnetic iron oxide (Fe3O4). Therefore, the crystal forms of iron oxide in PC-Fe/C-B were Fe3O4 and α-Fe2O3. It can be seen from Figure 2 that the characteristic peaks of α-Fe2O3 at 2θ = 24.50°, 40.85°, 49.53°, and 63.98° disappeared after As(V) adsorption, the main characteristic peaks for α-Fe2O3 at 2θ of 33.21° and 54.12° were significantly weakened and the characteristic peaks for Fe3O4 at 2θ of 29.98°, 43.34°, 57.50°, and 62.42° were strengthened, which indicated that the crystal form of iron oxide changed after adsorption. It was speculated that this may be due to the hydration reaction of some α-Fe2O3 dissolved in water to generate FeOOH or surface complexation reaction.” (Page 5, Lines 201~215)

Comment: 2. The conclusions are too general and should be refined.

Agreed with the reviewer's opinion and the sentence is rephrased, it now reads:

“In this study, PC-Fe/C-B material was prepared and the hierarchical porous microstructure of bamboo bio-templates was found to be accurately retained. The PC-Fe/C-B adsorbent was found to be effective for the removal of As (V) from wastewater under the following optimal experimental conditions: influent flow 5.136 mL/min, pH 3, As(V) concentration 20 mg/L, adsorbent particle size 0.149 mm, adsorption temperature 35°C, PC-Fe/C-B dose 0.5 g, and breakthrough time 50 min (184 BV) with qe,exp of 21.0 mg/g. The adsorption capacity of PC-Fe/C-B was higher than that of similar adsorbents reported in literature.

The surface characteristics of PC-Fe/C-B were simulated and analysed using the CD-MUSIC model with the PHREEQC program. The surface site density of PC-Fe/C-B was estimated as 2.5 sites/nm2, and the surface complexes ≡FeOAsO2OH, ≡Fe2O2AsOOH, ≡Fe2O2AsO2, and ≡FeOAsO2OH were dominant over a wide pH range.

These results showed that the CD-MUSIC model can be effectively coupled with the transport model using the program PHREEQC to simulate the reactive transportation of As(V) in the fixed-bed column and to predict the breakthrough curve for column adsorption. The fitting result obtained was superior to that of classic models such as the Thomas model and Yoon–Nelson model.” (Page 16, Lines 431~447)

Reviewer 4 Report

the manuscript can be suggested for publishing

Author Response

(The authors gave the same response as above.)

Reviewer 5 Report

The authors have provided the necessary revisions and the manuscript can be accepted.

Author Response

(The authors gave the same response as above.)
